



# Response of Late-Eocene warmth to incipient glaciation on Antarctica

Dennis H.A. Vermeulen[1], Michiel L.J. Baatsen[2], and Anna S. von der Heydt[2]

[1]Environmental Fluid Mechanics, Delft University of Technology, the Netherlands
[2]Institute for Marine and Atmospheric Research, Utrecht University, the Netherlands

**Correspondence:** Dennis H.A. Vermeulen (d.h.a.vermeulen@tudelft.nl)

**Abstract.** The Eocene-Oligocene Transition (EOT) is marked by a sudden $\delta^{18}O$ excursion occurring in two distinct phases, ~500 ky apart. These phases signal a shift from the warm Middle- to Late-Eocene greenhouse climate to cooler conditions, with global surface air temperatures decreasing by 3–5 °C and the emergence of the first continent-wide Antarctic Ice Sheet (AIS). While ice-sheet modelling suggests that ice sheet growth can be triggered by declining $pCO_2$, it still remains unclear how this transition has been initiated, in particular the first growth phase that seems to be related to oceanic and atmospheric cooling rather than ice sheet growth. Recent climate model simulations of the Late-Eocene show improved accuracy but depict climatic conditions that are not conducive to the survival of incipient ice sheets throughout the summer season. This study therefore examines whether it is plausible to develop ice sheets of sufficient scale to trigger the feedback mechanism(s) required to disrupt the atmospheric regime above the Antarctic continent during warm Late-Eocene summers and establish more favourable conditions for ice expansion. We thereby aim to assess the stability of an incipient AIS under varying radiative, orbital and cryospheric forcing. To do so, we evaluate Community Earth System Model 1.0.5 simulations, using a 38 Ma geo- and topographical reconstruction, considering different radiative (4 and 2 pre-industrial carbon; PIC) and orbital (present-day and low summer insolation) forcings. The climatic conditions prevailing during (the lead-up to) the EOT can be characterised as extremely seasonal and monsoonal, featuring a short yet intense summer period and contrasting cold winters — highly inhospitable to ice sheet growth for most of the continent, as limited snow accumulation is expected to survive the summer season. A narrow convergence zone with moist convection around the region where sub-cloud equivalent potential temperature is high is shown to exhibit a ring-like structure, advecting moist surface air advected from the Southern Ocean. This advection leads to high values of moist static energy and subsequent precipitation in these regions. To assess the influence of cryospheric forcing, we conducted another simulation, with regional, moderately-sized ice sheets imposed on the continent, to investigate their stability and influence on the atmospheric circulation. Regionally, these relatively small ice sheets respond strongly to radiative and orbital forcing, and demonstrate remarkably favourable self-sustaining and even expansion potential under 2 PIC and low summer insolation conditions. This emphasises a significant hysteresis effect for local and/or regional ice sheets on the Antarctic continent, suggesting the potential for a significant volume of ice on the Antarctic continent without an imminent full glaciation prior to the EOT.



## 1 Introduction

### 1.1 Eocene-Oligocene Transition

After the pinnacle of extreme warmth during the Early-Eocene Climatic Optimum (EECO; 53–51 Ma) greenhouse world, temperatures globally decreased towards the end of the Eocene (EO) (Anagnostou et al., 2016). The subsequent Eocene-Oligocene Transition (EOT; 34.44–33.65 Ma) shows a sudden (300–500 ky) oxygen isotope ($\delta^{18}$O) excursion in deep-sea benthic foraminifera (Hutchinson et al., 2021; Premoli Silva and Graham Jenkins, 1993). This excursion is commonly interpreted as the onset of the first continent-wide Antarctic ice sheet (AIS) (Lauretano et al., 2021; Carter et al., 2017; Scher et al., 2014). This sudden ice sheet growth on Antarctica led the planet to its present-day icehouse state (Pandey et al., 2021) and is — apart from this $\delta^{18}$O excursion — commonly implied by: 1. the deposition of ice-rafted debris around the Antarctic continental margins (Scher et al., 2011; Zachos et al., 1992) and glacial diamictites on the Antarctic Peninsula (Ivany et al., 2006); 2. the documentation of an extensive level fall in Antarctic coastal sediments, indicating large-scale ice growth on the continent (Katz et al., 2008; Stocchi et al., 2013); and 3. the shift from a chemical to physical weathering regime on Antarctica (Robert and Kennett, 1997), evident in mineralogical (Passchier et al., 2013) and geochemical (Houben et al., 2013) changes. Additionally, there are noticeable indications of decreasing sea surface temperatures (Liu et al., 2009) and changing ocean circulation around Antarctica (Coxall et al., 2018) during the EOT.

The aforementioned $\delta^{18}$O excursion occurred in two distinct phases, seperated by approximately 500 ky: during the first phase (precursor event; 34.15±0.04 Ma) a widespread cooling is evident without large-scale ice build-up (Scher et al., 2011; Katz et al., 2008), whereas during the second phase (Earliest Oligocene oxygen Isotope Step (EOIS); 33.65±0.04 Ma) major glaciation occurred without extensive cooling (Hutchinson et al., 2021) — whereupon $\delta^{18}$O rebounded to more balanced values (Passchier et al., 2017; Galeotti et al., 2016; Basak and Martin, 2013).

### 1.2 Inception of the Antarctic ice sheet

During the lead-up to the EOT two major changes have occurred, both of which have been hypothesised as potential causes for the onset of Antarctic glaciation. The first involves continental reconfiguration, which led to the opening of both Drake Passage (DP) between Antarctica and South America and the Tasmanian Gateway (TG) between Antarctica and Australia. The resulting gateway hypothesis suggests that the opening of these passages led to: 1. a comprehensive reorganisation of ocean currents around Antarctica; 2. the establishment of the Antarctic Circumpolar Current (ACC); and 3. eventual cooling of the Antarctic region. Although this hypothesis finds support in (deep) marine proxies (Bijl et al., 2013) and early numerical modelling (e.g., Kennett, 1977), it has faced criticism because of three primary reasons: the process of gateway opening occurs over longer timescales (millions of years) compared to the abrupt onset of the first AIS (Coxall et al., 2005); the timing of the opening of these gateways is poorly established, and does not seem to align with the timing of the glaciation (Wellner and Anderson, 2013); and modelling studies indicate that the opening of DP and TG has minimal to no impact on the formation of a continental Antarctic ice sheet (Pollard and DeConto, 2005; DeConto and Pollard, 2003).





(Marine) proxies such as boron isotope measurements (Pearson et al., 2009) and Mg/Ca ratios in benthic foraminifera (Katz et al., 2008) reveal a downward trend in atmospheric $p$CO$_2$ levels throughout the Cenozoic era, decreasing from 1400±470 ppm during the EECO to 750±50 ppm during the EOT (Hutchinson et al., 2021; Anagnostou et al., 2016). This reduction forms the

basis of the second hypothesis explaining the emergence of the AIS, as lower $p$CO$_2$ levels could create favourable conditions for ice growth. This viewpoint is further substantiated by data derived from terrestrial proxies and pollen analysis from regions proximal to Antarctica, e.g., by examining bacterial lipids in South Australia (Lauretano et al., 2021) and analysing $\delta$D in Patagonian volcanic glass (Colwyn and Hren, 2019). However, this hypothesis is also considered to have limitations, especially because the threshold of (model-derived) $p$CO$_2$ required to trigger glaciation seems to be heavily dependent on the specific

model used, often significantly lower than what is indicated by proxy data (Goldner et al., 2014).

To reconstruct the growth pattern of the AIS, however, we require proxy data regarding EOT conditions of the Antarctic continent itself. Consequently, other methodologies to investigate landscape formation are crucial, such as seismic reflection (Gulick et al., 2017), radar imagery (Rose et al., 2013; Bo et al., 2009) or digital elevation models (Barr et al., 2022). These methods reveal that ice sheet expansion likely commenced with the emergence of small, high-elevation mountain glaciers

within the interior mountain ranges of the continent (Rose et al., 2013; Wilson et al., 2013). These mountain glaciers contributed to the development of larger valley glaciers during a subsequent second phase, which then extended towards the continental margins (Bo et al., 2009), encompassing Sabrina Coast, Prydz Bay and the Weddell Sea (Gulick et al., 2017). The precise timing of these phases, however, remains unclear, and there are recent indications that high-elevation glaciers may have existed well before the EOT due to the occurrence of not only one but multiple EOT precursor events (Barr et al., 2022; Pandey et al., 2021).

Subsequently, a period of "flickering" transition, marked by numerous glaciation events, persisted until approximately 32.8 Ma, when a stable AIS had definitively formed, encompassing a size 70–110% of its present-day size and volume (Hutchinson et al., 2021).

### 1.3    Climate and ice sheet modelling across the EOT

When considering the $p$CO$_2$ drawdown hypothesis and the timing of glaciation phases, the question remains regarding the

degree to which declining $p$CO$_2$ influenced the onset of Antarctic glaciation. In this context, general circulation models (GCMs) are utilised alongside (offline) ice sheet models (ISMs). In recent years, these GCMs have exhibited a growing ability to simulate snapshot climatic conditions that increasingly align with available proxy data. The primary contribution of these model studies lies in establishing a $p$CO$_2$ threshold that functions as tipping point, beyond which continent-wide glaciation becomes inevitable (Tigchelaar et al., 2011). While this threshold has been previously set at 2.7±0.7 PIC (pre-industrial carbon, where

1 PIC comprises 280 ppm $p$CO$_2$ and 671 ppb $p$CH$_4$) (DeConto et al., 2008; Gasson et al., 2014; Ladant et al., 2014), it appears to be significantly model-dependent and more recent studies struggle to establish a consensus on a universal value (Kennedy-Asser et al., 2020, 2019; Hutchinson et al., 2018). This variability can be attributed — apart from the GCM and/or ISM used — to differences in initial and boundary conditions, primarily concerning: 1. the prescribed Antarctic topo- and geography along with the resultant atmospheric lapse rate; 2. the ab- or presence of an AIS; and 3. $p$CO$_2$. Insolation, driven by alterations

in orbital parameters, seems to exert a secondary influence by determining the timing of glaciation, specifically in relation





to cool(er) Southern Hemisphere summers; however, this effect is contingent upon $pCO_2$ surpassing the threshold (DeConto et al., 2008). Once continental glaciation takes hold, height-mass balance and ice-albedo feedbacks establish a hysteresis effect, rendering the AIS resistant to warmer climatic periods driven by less favourable orbital parameters (Van Breedam et al., 2022; DeConto and Pollard, 2003).

A limitation of older, low-resolution, model studies lies in their depiction of an unrealistic Middle- to Late-Eocene climate, falling short in reconstructing the relatively warm climate from the Bartonian to Priabonian epochs. Recent simulations conducted at higher resolutions, however, have shown improved accuracy (Lunt et al., 2021, 2017). Nevertheless, striving to align these Late-Eocene simulations more closely with proxy data, a paradox emerges: these simulations depict monsoonal climatic conditions above the Antarctic continent (Baatsen et al., 2024), that are not conducive to the survival of incipient ice sheets

throughout the summer season. This study will therefore examine whether — within the constrained potential for (limited) ice growth in the Late-Eocene Antarctic climate — it is plausible to develop ice sheets of sufficient scale to trigger the feedback mechanism required to disrupt the atmospheric regime above the Antarctic continent during the warm Eocene and establish more favourable conditions for ice expansion. To do so, we describe the simulations used as well as the tools we used to analyse both the stability of these ice sheets and the atmospheric regime under which they develop in Sect. 2. In Sect. 3 we

will present a characterisation of the atmospheric regimes of the different model scenarios considered and describe the climatic consequences of those regimes on the stability of incipient ice sheets. We discuss our results regarding atmospheric regimes and conditions for sustainability of (small) imposed ice sheets in Sect. 4. Finally, we will present our main conclusions in Sect. 5.

## 2   Methods

### 2.1   Model set-up and simulation scenarios


All model simulations were executed using the Community Earth System Model (CESM), version 1.0.5, developed at the American National Center for Atmospheric Research. This GCM comprises four distinct components: Community Atmosphere Model 4 (Neale et al., 2013), configured with a horizontal resolution of 2.5°×1.9° and encompassing 26 vertical hybrid-sigma levels extending up to 2 hPa; Parallel Ocean Program 2 (Smith et al., 2010), configured with a horizontal resolution

of 1.25°×0.9° and encompassing 60 vertical layers ranging in thickness from 10 m to 250 m; Community Land Model 4 (Lawrence et al., 2011), in which all anthropogenic influences have been eliminated, employed in a static configuration; and Community Ice Code 4 (Hunke et al., 2015), constrained to generate sea ice only when ocean water temperature falls below 1.8 °C.

   Three of four model simulations (Table 1) discussed here were conducted by Baatsen et al. (2020), all using 38 Ma boundary

conditions: 1. incorporating 4 PIC (4.69 pre-industrial (PI) $pCO_2$ equivalent) $pCO_2$ forcing, to simulate the warm Middle-Eocene climate (4PIC); 2. incorporating 2 PIC (2.15 PI $pCO_2$ equivalent) $pCO_2$ forcing, to replicate a cooling trend towards the EOT during the Late-Eocene (2PIC/s); and 3. incorporating 2 PIC $pCO_2$ levels plus orbital parameters that induce minimal summer insolation at southern high latitudes (2PIC/l), to prompt favourable conditions for ice growth. For more information





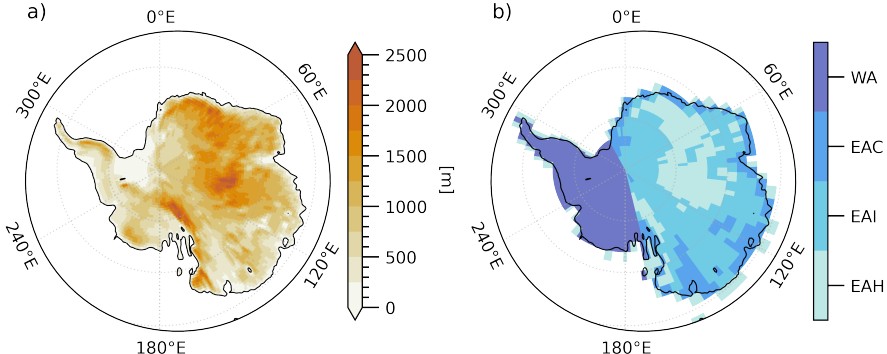

**Figure 1.** Model topography shown in panel a). Regional differentiation of the Antarctic continent is shown in panel b). West-Antarctica (WA) includes all area between 166°E and 330°E; East-Antarctica coast (EAC) encompasses all other area below 500 m; East-Antarctica interior (EAI) covers all other area between 500 m and 1300 m; and East-Antarctica highlands (EAH) comprises all area above these elevation values.

**Table 1.** Model parameters for the four distinct scenarios modelled by Baatsen et al. (2020). $e$ represents eccentricity, $\varepsilon$ obliquity, $\tilde{\omega}$ precession and $F_s$ the solar constant.

|  | **4PIC** | **2PIC/s** | **2PIC/l** | **2PIC/li** |
|---|---|---|---|---|
| $e$ [–] | 0 | 0 | 0.064 | 0.064 |
| $\varepsilon$ [°C] | 23.44 | 23.44 | 22.315 | 23.315 |
| $\tilde{\omega}$ [°C] | n/a | n/a | 1.591 | 1.591 |
| $F_s$ [W m$^{-2}$] | 1361.27 | 1361.27 | 1361.27 | 1361.27 |
| $p$CO$_2$ [ppm] | 1120 | 560 | 560 | 560 |
| $p$CH$_4$ [ppb] | 2684 | 1342 | 1342 | 1342 |

about specific model parameters and initialisation, see Baatsen et al. (2020). The final 100 yr of each model run are used in this

study to generate a representative climatology, distinguishing four different Antarctic regions (Fig. 1). For clarity, greenhouse gas forcings will be indicated as radiative forcing in the text below; insolation forcings will be indicated as orbital forcing.

We used 2PIC/s to initialise ice sheet model IMAUICE (Berends et al., 2022), an ISM that employs the shallow-ice approximation (SIA) and incorporates a modified topography from Wilson et al. (2012), projected onto an Antarctic Polar Stereographic grid with a horizontal resolution of 40×40 km. IMAUICE was run until equilibrium was achieved after 30 ky. The

model output was regridded to align with the CESM grid using the GPlates software package (Boyden et al., 2011) and a set of reconstructed plate trajectories (van Hinsbergen et al., 2015; Seton et al., 2012). Surface mass balances (SMBs) were computed using the insolation-temperature method (de Boer et al., 2014). Various ISM simulations were conducted with different lapse rates and orbital, melt, and stress parameters; however, no combination resulted in large-scale ice growth. Nevertheless,





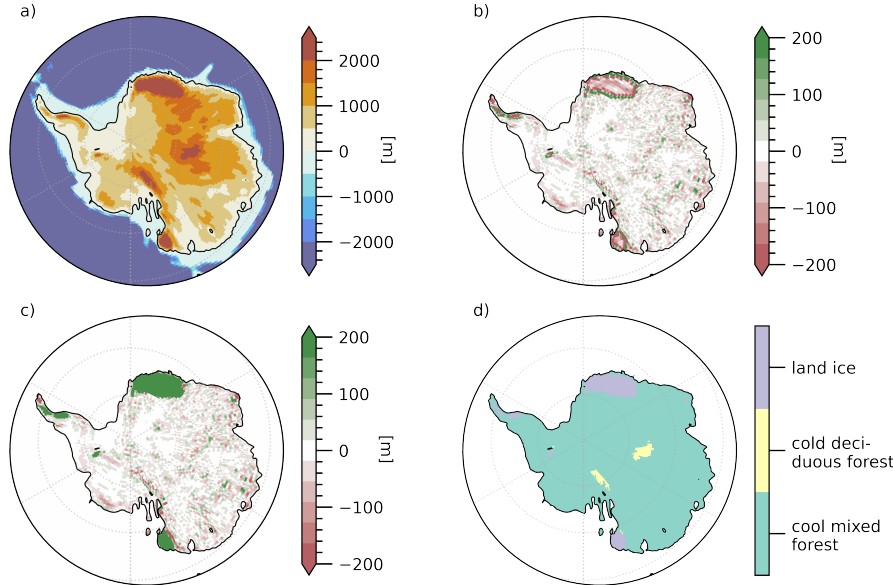

**Figure 2.** Ice grid transformed into a vegetation mask for input to CESM1.0.5. Panel a) displays Antarctic surface topography with w1m05 ice sheets, smoothed using a 3×3 kernel. Panel b) shows the difference between the effect of the grid smoothing, i.e., the difference between the unsmoothed Antarctic surface topography with w1m05 ice sheets and the smoothed topography. Panel c) illustrates the difference between the original (ice-free) input for 2PIC/l and the smoothed topography. Panel d) depicts the resulting vegetation mask.

inducing a cooling of mean summer (DJF) $T_\text{s}$ by 5 °C did trigger ice sheet growth (w1m05; Fig. 2), leading to the development

of moderately-sized, stable ice sheets in Dronning Maud Land, Oates Land and the West-Antarctic Peninsula.

In an effort to explore the potential for the development of an incipient ice sheet under the most favourable climatic conditions, we initiated for this study an additional CESM simulation to the above-mentioned three, branched off of the end of 2PIC/l. This simulation, 2PIC/li, is conducted over a 400 yr period and incorporates a vegetation/land mask that accounts for land ice (Fig. 2). The ice sheet topography in this land mask is based on the above-mentioned w1m05 simulation, since the

resulting ice sheets in w1m05 are stable in form and the reduction in summer $T_\text{s}$ between 2PIC/s and 2PIC/l due to favourable orbital forcing is also ~5 °C. Since our primary interest lies in atmospheric output and the only alteration in 2PIC/li is the sudden imposition of several moderately-sized regional ice sheets, stable outcomes were achieved within a relatively short simulation time of ~75 yr, no spin-up necessary (not shown here). Nevertheless, to avoid any potential complications, we adopted 100 yr climatologies from the interval between 300–400 yr simulation time.

## 2.2 Atmospheric metrics

We differentiate three variables to compare the atmospheric regime above the Antarctic continent between the different scenarios mentioned above, focusing on the examination of moisture and energetic characteristics of the atmosphere:





$$Q_{\text{net}} = \text{SW}_{\text{s}} - \text{SW}_{\text{ToA}} + \text{LW}_{\text{s}} - \text{LW}_{\text{ToA}} + \text{LHF} + \text{SHF} \tag{1}$$

with $Q_{\text{net}}$ [W m$^{-2}$] being the net radiation balance, cf. Boos and Storelvmo (2016), where SW represent shortwave radiation,
LW longwave radiation, LHF latent heat flux, SHF sensible heat flux, ToA the top of the atmosphere, and s the surface.

$$\theta_{\text{eb}} = T \left( \frac{p_{\text{s}}}{p_{\text{d}}} \right)^{\frac{R_{\text{d}}}{c_{\text{d}} + c_{\text{l}} q}} H^{\frac{-q R_{\text{v}}}{c_{\text{d}} + c_{\text{l}} q}} \exp^{\frac{L_{\text{v}} q}{(c_{\text{d}} + c_{\text{l}} q) T}} \tag{2}$$

with $\theta_{\text{eb}}$ [K] being ~980 hPa sub-cloud equivalent potential temperature, cf. Hurley and Boos (2013), where $p_{\text{s}}$ represents
surface pressure and $p_{\text{d}}$ dry air pressure at the level of $T$, $R_{\text{d}}$ the gas constant for dry air and $R_{\text{v}}$ for water vapour, $c_{\text{d}}$ specific
heat at constant pressure for dry air and $c_{\text{l}}$ for liquid water, and $H$ relative and $q$ specific humidity.

$$\text{MSE} = -\frac{1}{g} \int_{p_{\text{s}}}^{200} [c_{\text{p}} T + gz + L_{\text{v}} q] \, dp \tag{3}$$

with MSE being vertically-integrated (from the surface to the tropopause at 200 hPa) moist static energy [GJ m$^{-2}$], cf. Smyth
and Ming (2020) and Hill et al. (2017), where $L_{\text{v}}$ represents latent heat of vaporisation, $c_{\text{p}}$ specific heat at constant pressure
for water vapour, and $z$ the air parcel's height.

$Q_{\text{net}}$ aids in analysing the atmosphere's radiative state. Employing a convective quasi-equilibrium framework, $\theta_{\text{eb}}$ offers in
turn a simplified depiction of the vertical tropospheric structure as a single near-surface variable. Although this simplification
undoubtedly overlooks complexities, it facilitates the analysis of the atmosphere's convective state. A similar rationale applies
to MSE, which remains conserved during adiabatic ascent, and helps analysing the atmosphere's energetic state.

### 2.3 Ice sheet stability

In order to evaluate the potential sustainability of regional ice sheets under 2PIC/li climatic conditions, we use four different
methods to calculate surface mass balance (all in units of m yr$^{-1}$). The first method is based on the surface energy balance,
cf. van den Broeke et al. (2011) (SMB$_1$). The three other methods are based on the amount of positive degree days (PDDs).
PDDs serve as proxy for the available melt energy on days with a positive $T_{\text{s}}$ (Braithwaite, 1984). Assuming that all annual
precipitation falls as snow, this melt can be calculated in different ways: employing a fixed melt rate with PDD calculated as
the annual sum of $T_{\text{s}}$ exceeding 0 °C, cf. Heyman et al. (2013) (SMB$_2$); employing a fixed melt rate with PDD calculated using
a semi-empirical linear relationship that incorporates a $T_{\text{s}}$ variability term to account for missing information about synoptic
variability and the diurnal cycle, cf. Bauer and Ganopolski (2017) (SMB$_3$); and employing the aforementioned semi-empirical
linear relationship, but introducing separate melt rates for snow and ice melt (ice melt occurs when all annual precipitation —
in the form of snow — is melted away, and the ice sheet base becomes exposed) and considering a refreezing factor for the
nocturnal refreezing of snow, cf. Bauer and Ganopolski (2017) (SMB$_4$).





The equations governing these four methods are as follows:

$$\mathrm{SMB}_1 = -\frac{\mathrm{SW_s} + \mathrm{LW_s} + \mathrm{LHF} + \mathrm{SHF}}{c_i} \tag{4}$$

$$\mathrm{SMB}_2 = P_{\mathrm{ann}} - \alpha \Sigma T_s^+ \tag{5}$$

$$\mathrm{SMB}_3 = P_{\mathrm{ann}} - \alpha \mathrm{PDD} \tag{6}$$

$$\mathrm{SMB}_4 = \begin{cases} \alpha_i Q - P_{\mathrm{ann}} & Q < 0 \\ P_{\mathrm{ann}} & Q = 0 \\ \alpha_s (1 - r_s) Q - P_{\mathrm{ann}} & Q > 0 \end{cases} \tag{7}$$

with

$$\mathrm{PDD} = \int_{\Delta t} \left[ \frac{\sigma}{\sqrt{2\pi}} \exp^{-\frac{T_s^2}{2\sigma^2}} + \frac{T_s}{2} \mathrm{erfc}\left( -\frac{T}{\sqrt{2}\sigma} \right) \right] \mathrm{d}t$$

and with

$$Q = \frac{P_{\mathrm{ann}}}{\alpha_s (1 - r_s)} - \frac{\mathrm{PDD}}{\Delta t}$$

and where $c_i = 333.55$ kJ kg$^{-1}$ represents the latent heat of fusion of water; $\alpha = 4$ mm $°$C$^{-1}$ day$^{-1}$ represents a fixed melt rate, and $\Sigma T_s^+$ the annual sum of $T_s$ above 0 $°$C; where $\Delta t = 1$ yr represents the time step, $\sigma = 5$ $°$C a fixed standard deviation of $T_s$ (representative for present-day ice sheets, cf. Bauer and Ganopolski (2017), and since no specific information is available

about empirical relationships for $T_s$ in the Eocene, we apply it in this context as well), and erfc$(x)$ the complementary error function; and where $\alpha_s = 3$ mm $°$C$^{-1}$ day$^{-1}$ represents the snow melt rate, $\alpha_i = 8$ mm $°$C$^{-1}$ day$^{-1}$ the ice melt rate, and $r_s = 0.3$ the refreezing factor. Furthermore, SMBs are calculated in such a way that ice melt only occurs over ice sheets, and all SMBs are defined positively for accumulation.

## 3   Results

### 3.1   Climatologies of Antarctica during the EOT

Figure 3 presents a climatology of 2 m surface temperature ($T_s$) and monthly precipitation ($P$) for all four model scenarios, distinguishing between the above-mentioned regions. $P$ displays the most variation in 4PIC, with high levels in December and





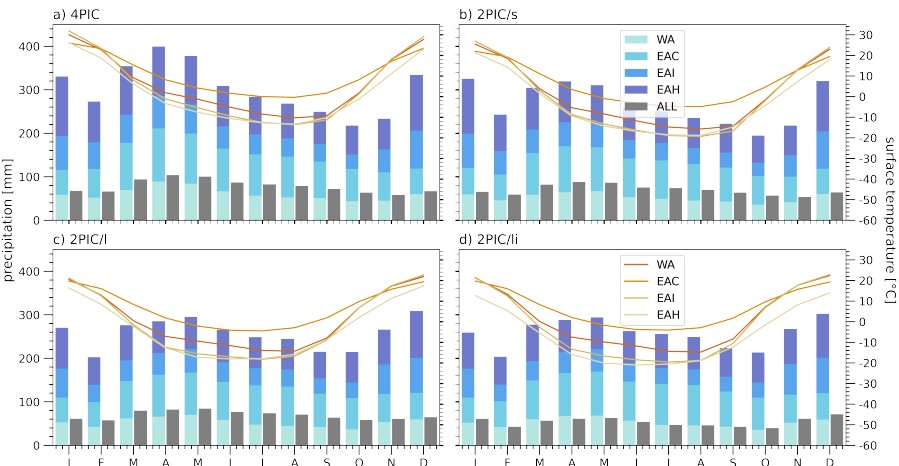

**Figure 3.** 100 yr climatology for different Antarctic regions. All four EO scenarios are depicted. Lines represent $T_s$; bars represent monthly $P$. Different colours indicate the different regions. The grey bars represent continent-averaged (i.e., not summed) monthly $P$.

January, followed by a sharp decrease in February, a substantial increase in autumn (MAM), and gradual drying from June to October. This variability is less pronounced in the other two scenarios, although DJ and MAM retain their relatively wet

periods. Among the EO scenarios, EAI emerges as the warmest but also driest region; EAC and EAH are the coldest areas, with EAC experiencing significant $P$ in autumn and winter, while EAH is notably wet during summer only.

Due to the rapid decrease in $T_s$ between January and February, we define summer for the remainder of this study as DJ (i.e., not DJF). Figure 4 then illustrates JJA- and DJ-mean and JJA-mean and DJ-mean anomalies of $T_s$, $P$, 850 hPa wind ($u_{850}$) and mean sea-level pressure ($p_s$). Additionally, Table A1 provides the regional mean and spatial standard deviation of these

variables. Summers are notably warm (reaching up to 35 °C), particularly in EAI, whereas winters are cold (ranging between -15 °C and -20 °C). Annual $T_s$ does not exhibit a consistent decrease with declining insolation, yet significant differences emerge in $T_{s,DJ}$ for all regions except EAC. Among all scenarios, EAC displays the weakest seasonality, characterised by cool summers and mild winters, while EAI shows the most pronounced contrast between the seasons. The seasonal variation in $P$ is also evident. In EAI, summer (~3 mm day$^{-1}$) is wetter than winter (~1.5 mm day$^{-1}$). In EAC, particularly along the

coastal regions of Dronning Maud Land and the areas more eastward, winters receive substantial precipitation (up to 8 mm day$^{-1}$). Conversely, in WA, there exists minimal disparity between $P_{DJ}$ and $P_{JJA}$. This seasonality decreases when radiative and orbital forcing are reduced.

Apart from the effects of heat and moisture, we observe ascending air above the centre of the continent during summer. This results in weak 10 m winds that move from the 50–100°E sector towards the South Pole, and from the pole either towards ~0°E

or the 180–250°E sector (see Fig. 1 for the longitude grid). However, $u_{850}$ remains low (1–2 m s$^{-1}$) and decreases somewhat when radiative forcing and summer insolation decrease. In winter, on the other hand, the situation is different: a high-pressure area with strongly descending air forms above the Antarctic interior, causing the meridional $p_s$ gradient to intensify. This





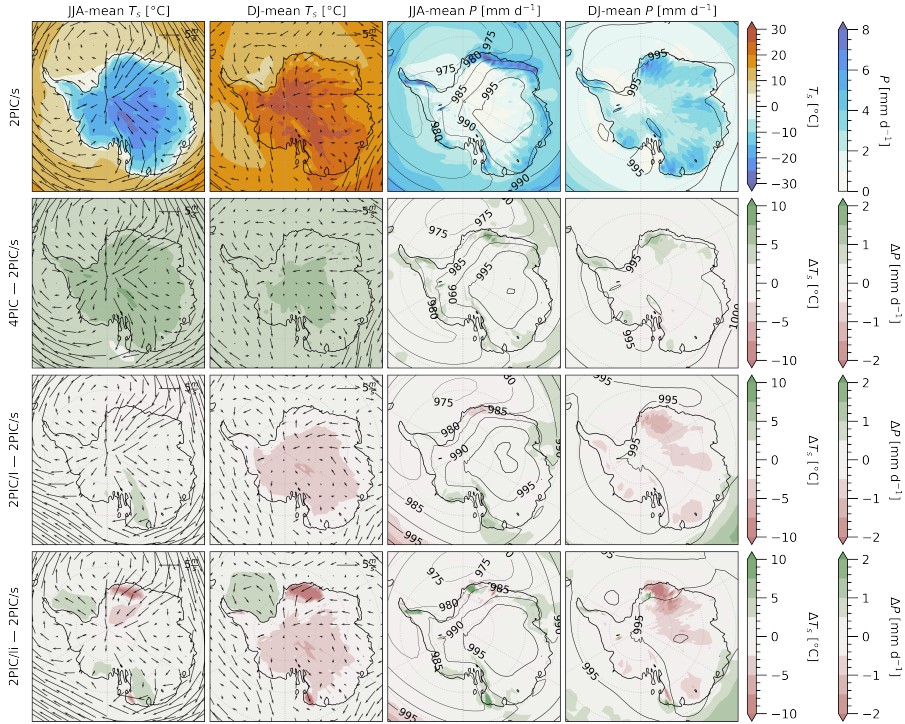

**Figure 4.** DJ- and JJA-means for 2PIC/s, and DJ- and JJA-mean $T_s$ and $P$ anomalies for the other three scenarios. Maps display $T_s$ (shading in columns 1 and 3), $P$ (shading in columns 2 and 4), $u_{850}$ (vectors in columns 1 and 3) and $p_s$ (contour lines in columns 2 and 4). The anomaly rows for $T_s$ and $P$ show $u_{850}$ and $p_s$ means, rather than anomalies.

results in a flow coming from the 0–70°E sector over the South Pole towards the 180–230°E sector. When taking orographic effects into account, this circulation leads to wet conditions on the coast of Dronning Maud Land and the eastern part of the

West-Antarctic Peninsula. This circulation pattern persists throughout all four scenarios.

A regional analysis of the spread between $T_s$ and $P$ reveals that decreasing radiative and orbital forcing predominantly effects $T_s$: average $P$ in these three scenarios remains relatively stable (~2.2 mm day$^{-1}$), while $T_s$ undergoes a significant reduction from ~28 °C to ~20 °C (Fig. 5). A similar trend is observed for WA (exhibiting nearly identical behaviour), EAI (although with a wider range in $P$) and EAC (with a smaller range in $T_s$). On the other hand, EAH shows a different regime,

as both $T_s$ and $P$ decrease, demonstrating a significant correlation between $T_s$ and $P$ with $r = 0.67$. For ALL, EAI and EAC, correlations are significant but weak, with $r = 0.22$, $r = 0.13$ and $r = -0.15$, respectively. WA does not show any significant correlation. Furthermore, we observe that the imposition of ice sheets only significantly influence $T_s$ and $P$ in EAH (becoming colder and drier, although mainly over the imposed ice sheets themselves), whereas EAI and WA appear to become slightly warmer and the continent-wide average is not influenced at all.

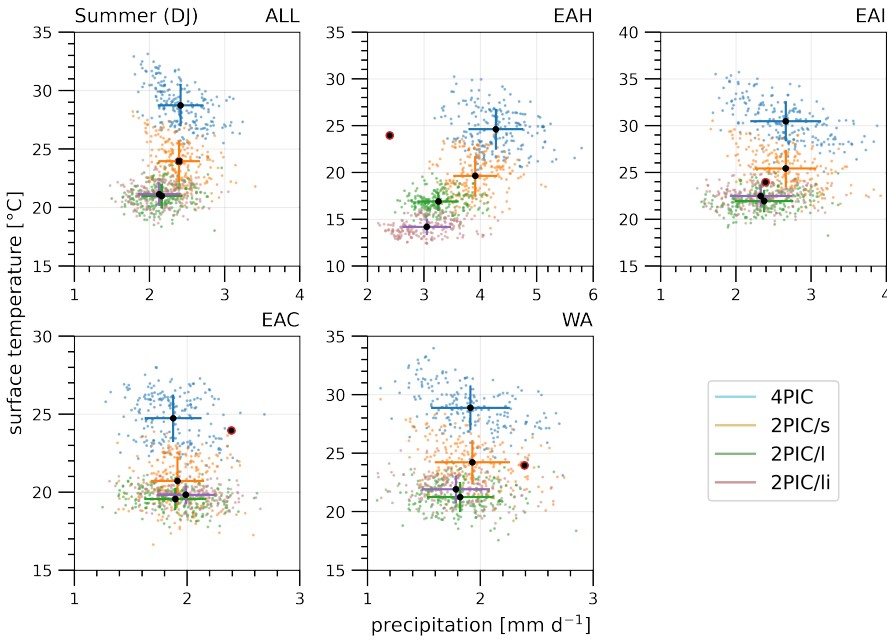

**Figure 5.** $T_s$ yearly variation plotted against their corresponding $P$ data point, categorised into an Antarctica-wide average and the four regions. The mean is depicted by a black marker, and the coloured lines indicate one (spatial) standard deviation. The x and y axes are not consistent along the panels; therefore a black dot with red border is plotted in every panel to mark the 2.5 mm day$^{-1}$ / 25 °C point.

### 3.2 Atmospheric regime during the warm Eocene


Heating of the EO Antarctic atmosphere initiates convection and subsequent precipitation, driven by a net radiation surplus. The influence of a low-insolation orbital configuration becomes evident when comparing $Q_{net}$ between 2PIC/s and 2PIC/l, revealing a $Q_{net}$ reduction of ~70% (Fig. 6) and Table A2). Among regions, WA and EAH receive the continent-average amount of summer radiation, while EAI receives more and EAC receives less, in all scenarios. However, there is significant

spatial variability within these regions, given the large standard deviation in average $Q_{net}$. Large amounts of radiation lead to elevated $T_s$. Notably, the substantial difference in orbital forcing between 2PIC/s and 2PIC/l does not yield a similar difference for $\theta_{eb}$. EAC consistently exhibits lower $\theta_{eb}$ in summer compared to EAH and EAI.

The attraction of moisture from the surrounding Southern Ocean yields high values for MSE across the southwest of Dronning Maud Land, as well as over Oates and Adélie Land (Fig. 6). Moisture tends to be most abundant in coastal regions

and progressively diminishes further inland in EAI and EAH, likely due to moisture raining out when ascending. This atmospheric configuration leads to convection of moist air over these areas, which inevitably results in precipitation. We previously discussed $P$, revealing the wettest conditions over Dronning Maud, Oates, Adélie and George V Land as well as in the Gamburtsevs — elevated areas relative close to the Antarctic coast. Especially Dronning Maud Land experiences substantial $P$ during summer. Notably, a reduction in available MSE becomes evident with decreasing $pCO_2$ and insolation, declining with

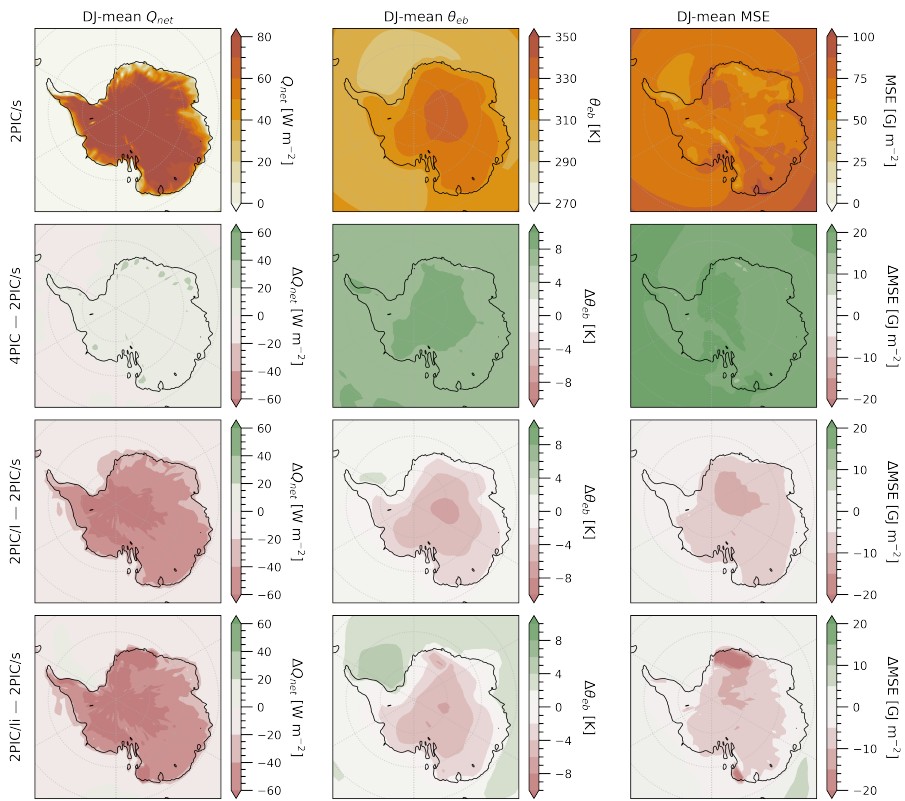

**Figure 6.** DJ-mean for 2PIC/s, and DJ-mean $Q_{net}$ (left panels), $\theta_{eb}$ (middle panels) and MSE (right panels) anomalies for the other three scenarios. The contours have different scales for each variable; however, they are normalised to effectively display the range of possible values.

approximately 30%. While Oates and Adélie Land maintain relatively high MSE, even in 2PIC/l, MSE decreases over Dronning Maud Land in 2PIC/l.

### 3.3 Effects of imposed ice sheets

The introduction of moderately-sized, regional ice sheets on the Antarctic continent under favourable orbital forcing leads to a slightly drier climate, accompanied by temperatures comparable to those in 2PIC/l (Fig. 3). Throughout the year, $p_s$ remains marginally higher in 2PIC/li compared to 2PIC/l. The EAH region displays the most significant reduction in $T_s$ and $P$, due to the inclusion of ice sheet areas within its bounds (Fig. 4). While EAI and WA experience a modest increase in $T_s$ (likely attributed to the same factor — the exclusion of areas from EAI that become ice sheets leads to a shift in weightage towards warmer grid cells, increasing the area mean) EAC shows minimal differences. The continent-wide $T_s$ mean for 2PIC/l and 2PIC/li exhibit nearly equivalent values. Notably, the largest standard deviations are observed for $T_{s,JJA}$, implying more pronounced regional anomalies during winter than during summer (Table A1).





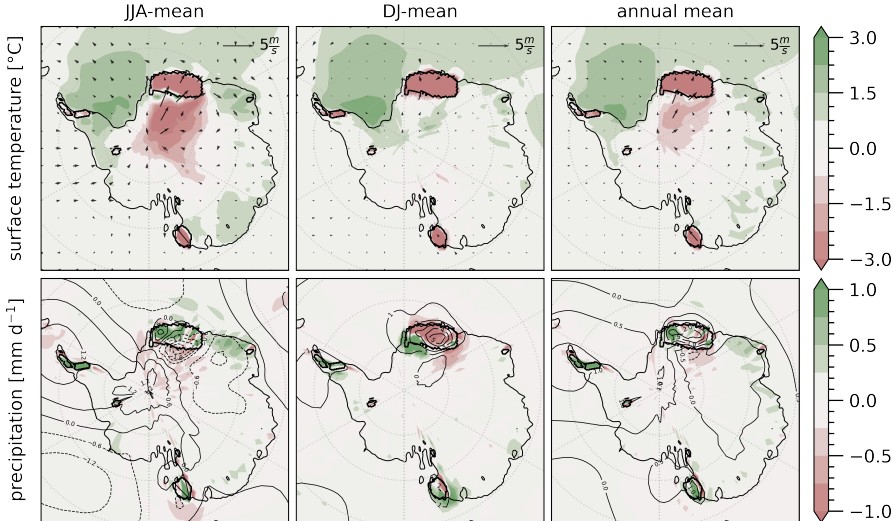

**Figure 7.** DJ- and JJA-mean anomalies between 2PIC/li and 2PIC/l climatologies. Maps display $T_\mathrm{s}$ (left panels), $P$ (middle panels), PR (right panels), $u_{850}$ (vectors in left panels) and $p_\mathrm{s}$ (contours in middle panels) anomalies.

The regions where the imposed ice sheets are present consistently exhibit lower temperatures in both summer and winter (Fig. 7). Over the ice sheet areas, conditions become drier in summer and wetter in winter. Conversely, surrounding areas (situated south of the Dronning Maud Land ice sheet and west of the ice sheets on Oates Land and the Peninsula) experience increased wetness in summer and reduced wetness in winter. Notably, the Weddell Sea, as well as Enderby and Kemp Land,
demonstrate higher $T_{\mathrm{s,JJA}}$, with the latter showing only slight warming in summer. On the other hand, EAH and EAI experience substantial cooling during winter, while the Transantarctic Mountains display a minor cooling trend in summer. The decline in $T_\mathrm{s}$ to the south of Dronning Maud Land in winter appears to be linked to a significant shift in wind patterns, marked by a reduced onshore component of the wind which leads to a reduced impact of mild maritime air on the region around the South Pole. This change is associated with a slight increase in $p_\mathrm{s}$ over WA and a slight decrease over EAI and EAC, yielding a
weakened airflow across the continent from Dronning Maud Land to Marie Byrd Land, albeit with modest anomalies. During summer, $p_\mathrm{s}$ significantly increases over the Dronning Maud Land ice sheet, along with a slight increase over the rest of the continent.

The warm and moist atmospheric circulation during Antarctic summer remains largely unchanged on a large scale when regional ice sheets are imposed (Fig. 8). However, notable regional anomalies emerge, especially over and near the ice sheets,
affecting all three atmospheric metrics. The high albedo of the ice sheets leads to reduced $Q_\mathrm{net}$, resulting in lower $\theta_\mathrm{eb}$ values within the ice sheet centres. When ice sheets are small, such as on the Peninsula, there is little effect at all. Moreover, the presence of ice sheets leads to decreased moisture content and MSE. A distinct feature is observed in the Weddell Sea area, where higher values of $Q_\mathrm{net}$ and $\theta_\mathrm{eb}$ along with slightly higher MSE levels are noted. Additionally, the coastal regions of




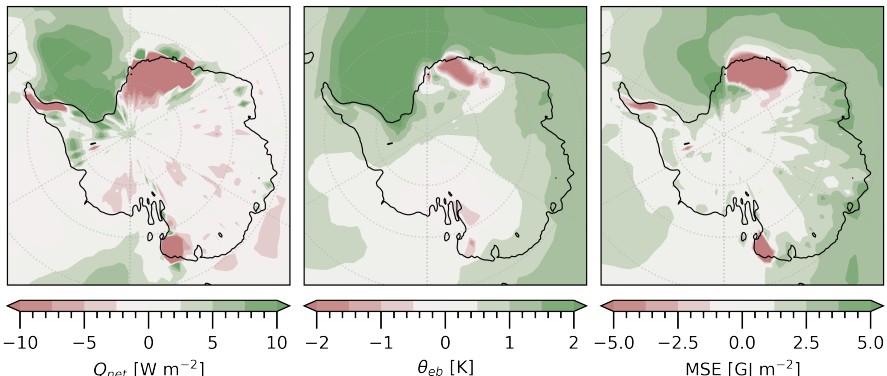

**Figure 8.** DJ-mean $Q_{net}$, $\theta_{eb}$ and MSE anomalies between 2PIC/li and 2PIC/l. The contours have different scales for each variable; however, they are normalised to effectively display the range of possible values.

East-Antarctica exhibit slightly higher $\theta_{eb}$ along with increased MSE values. Note also that $Q_{net}$ slightly decreases over much
of the continent, while both $\theta_{eb}$ and MSE seem to slightly increase.

In the vertical, significant anomalies are observed over the ice sheet regions in terms of meridional-mean (68°S–74°S) latent heating rate $L$, moisture flux $vq$ and meridional wind $v$ (Fig. 9). In the western sector of the Dronning Maud Land ice sheet, we observe greater $L$ anomalies, while the eastern part exhibits reduced $L$, particularly at higher altitudes. Over the central part, a more pronounced southward $vq$ is noted, in contrast to the flanks where northward transport prevails. Conversely, the ice
sheet on Oates Land shows an opposite trend, featuring northward transport in the central region and southward transport along the flanks; furthermore, here we only observe a positive $L$ anomaly. Lastly, winds tend to be directed more towards the south above the ice sheet surfaces, while on the flanks at height they exhibit an inclination towards the north. The atmosphere above non-ice-covered surfaces displays only limited anomalies, whereas at higher altitudes only the meridional wind component appears to be influenced.

With respect to SMBs in 2PIC/li, all methods yield a (strongly) positive SMB over the ice sheet areas, although SMB2 only shows positive values over the central, higher parts of the ice sheets (Figs. 10, right panels, and A1). SMB1 shows less strong positive values, but it covers a larger area where SMB is positive, indicating more potential for ice sheet expansion. Incorporating a variability term for $T_s$ in calculating PDD (i.e., comparing SMB2 and SMB3) results in a slightly larger area with positive SMB; differentiating between different melt factors for snow and ice (SMB4) yields an even larger area for ice
sheet expansion. This last method also indicates the potential for some ice sheet growth in the high areas of the Transantarctic Mountains and the Gamburtsevs, as well as towards the coastal areas of Enderby and Kemp Land. The integrated SMB over the entire continent is -2.68 km yr$^{-1}$ for SMB1, 3.61 km yr$^{-1}$ for SMB2, 1.47 km yr$^{-1}$ for SMB3, and 1.45 km yr$^{-1}$ for SMB4. When comparing SMB1 and SMB4 for the 2PIC/s and 2PIC/l scenarios (Fig. 10, left and middle panels), the same areas for ice sheet expansion become evident. We observe slightly larger areas with positive SMB when comparing 2PIC/l with 2PIC/s, but



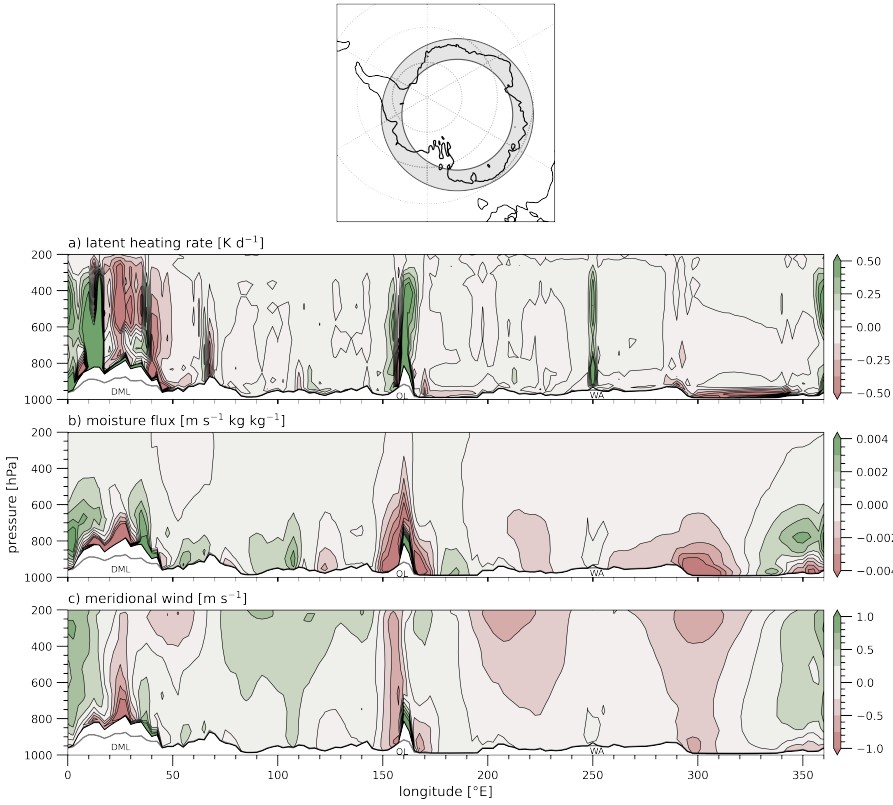

**Figure 9.** Meridional-mean DJ anomalies between 2PIC/li and 2PIC/l in $L$ (upper panel), $vq$ (middle panel), and $v$ (lower panel). The circle above shows the area over which the anomalies are averaged (i.e., between 68°S and 74°S, with the centre of the Eocene Antarctic continent as 90°S). Red (blue) contours indicate positive (negative) anomalies for all three variables. The x-axis shows longitude; due to the slightly off-centre placement of the circle around the South Pole, longitude slightly deviates from the circle angle. The thick grey line denotes 2PIC/l topography, while the thick black line denotes 2PIC/li topography. DML denotes the Dronning Maud Land ice sheet, OL the ice sheet from Oates Land and WA the ice sheet on the Peninsula.

differences are small and largely regional. Nevertheless, integrated SMB over the entire continent for both 2PIC/s and 2PIC/l remains strongly negative.

## 4   Discussion

### 4.1   Climatologies of Antarctica during the EOT

The 38 Ma Antarctic climate with 4PIC $p$CO$_2$, representing warm periods before the EOT, can be characterised as highly
seasonal, featuring very warm summers and cold winters. Notably, Dronning Maud, Oates, Adélie and George V Land exhibit wet conditions during summer, while a narrow coastal zone between 80°W and 80°E experiences excessive precipitation in





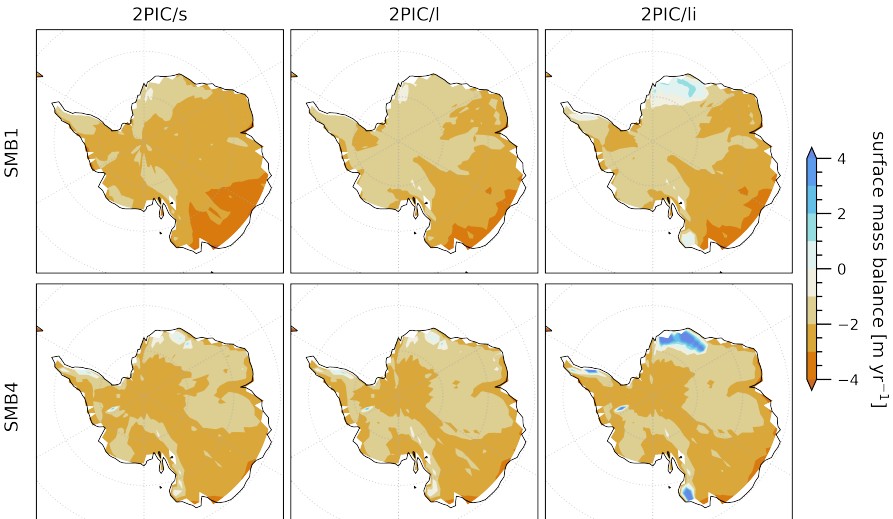

**Figure 10.** Surface mass balances of imposed regional ice sheets for 2PIC/li (right panels), calculated using two different methods: (1) one based on the surface energy balance (SMB1) and (2) a PDD method including a variability term plus using a refreezing factor and separate melt rates for snow and ice (SMB4). Both methods are also used to consider SMBs for 2PIC/s and 2PIC/l (left and middle panels). For results for SMB2 and SMB3, see Fig. A1.

winter (Fig. 4). This pronounced seasonality in $T_s$ weakens slightly when $p$CO$_2$ is reduced to 2 PIC, and further diminishes with alterations in orbital parameters that result in reduced summer insolation. However, the most prominent distinction among the various scenarios is a decline in mean $T_s$: ~7 °C for summer and ~5 °C for winter (Table A1). Furthermore, orbital forcing

exerts a more substantial influence than radiative forcing: the interior of Dronning Maud and Oates Land, both characterised as wet regions, exhibit high sensitivity to decreased summer insolation (resulting in reduced precipitation). In contrast, Wilkes Land and the coastal areas of Dronning Maud Land receive increased precipitation with higher summer insolation. This climate is persistent over time, devoid of periodic fluctuations.

The present-day (continent-wide) AIS creates a thick, dense layer of air in which cold air parcels become trapped, unable to

ascend (Hoskins, 1991). As a result, a robust anticyclone forms above the continent's surface during both summer and winter. This atmospheric configuration is accompanied by subsiding air, coupled with the occurrence of katabatic winds along the continent's periphery, contributing to cold conditions and limited moisture transport from the Southern Ocean towards the continent, thereby resulting in markedly low $P$ (Baatsen, 2018). In contrast, these EO simulations reveal a robust meridional flow that facilitates significant moisture transport towards Antarctica. This flow emerges due to the absence of such a thick,

dense air layer and the ensuing diminished meridional temperature gradient, but only in summer. This amplified warmth, coupled with markedly higher $T_s$, generates substantially increased moisture levels over the continent. During EO winter, this configuration results in a cyclonic polar vortex situated over the Weddell Sea, accompanied by an anticyclone positioned above EAI and EAH. These combined features engender a meridional flow that traverses the continent (first column in Fig.





4). This is a crucial part of the Antarctic Eocene climate, as the circulation pattern basically redirects the storm track (which
is already close to the continent) inland resulting in very high winter precipitation over Dronning Maud Land, but also over
West-Antarctica. In summer, a persistent cyclone forms, characterised by relatively weaker winds over EAI and EAH. Within
this cyclonic system ascending air is observed, providing conducive conditions for the initiation of convection.

We compare the results discussed above with findings from five other GCMs (figures not shown here), evaluated within the
context of the Eocene Model Intercomparison Project (EoMIP; Lunt et al. (2012)) and the Deep-Time Model Intercomparison
Project (DeepMIP; Lunt et al. (2021, 2017)). While the time period under investigation in both projects is focused on the EECO
(~50 Ma), boundary conditions are akin to those employed in this paper (Table A3). Various paleogeographical reconstructions
have been utilised, but especially for Antarctica, South America, Australia, Drake Passage and the Tasmanian Gateway only
minor changes occurred between 55 Ma and 38 Ma.

We see that CESM1.0.5, the model utilised throughout this study, exhibits relatively warm conditions; similarly, high-
resolution models similar to our study show EAI $T_s$ within the range of 25–30 °C, with cooler temperatures (15–20 °C) in
EAC. Strikingly, CESM1.2 displays a distinct pattern, portraying less warm $T_s$ and cold air over the Gamburtsevs. Regarding
$p_s$, high pressure dominates over the continent in all models, although magnitudes vary across. CESM1.0.5. in the 4PIC scenario
is very wet for DJ: only IPSL comes up with a similar $P$ pattern over Dronning Maud Land but its magnitude is smaller than
for 4PIC, whereas CCSM3 and HadCM3 show only a moderate signal in EAI. CESM1.2 and NorESM show barely any signal
at all. Notably, all models except CCSM3 project relatively moist conditions during summer over Dronning Maud, Wilkes and
Oates Land and the Peninsula, but other seasons demonstrate even higher levels of $P$. The influence of the atmospheric model
used (e.g., CAM4 in CESM1.0.5 and CAM5 in CESM1.2 and in other DeepMIP models) does not appear to make a significant
difference. Although the climate sensitivity of CAM5 appears to be higher (Zhu et al. (2019)), CAM4 also simulates high
temperatures at high latitudes. DeepMIP simulations using CAM5 do not show significantly different results compared to the
CAM4 configuration shown here. Hence, resolution and topo-/geography seem to be more important in this context than the
choice of atmospheric model.

## 4.2  Atmospheric regime during the warm Eocene

The circulation during EO Antarctic summers is too remote to be directly connected to the overturning of the (sub)tropical
atmosphere. Examining the continent's radiative balance, (very) high $Q_{\mathrm{net,DJ}}$ values for 4PIC and 2PIC/s are evident, primarily
due to the continuous austral summer sun (Table A2). The influence of orbital parameters is also apparent, with low insolation
parameters in 2PIC/l leading to a large reduction in $Q_{\mathrm{net}}$. This surplus of available energy initiates a range of interconnected
processes, beginning with an increase in tropospheric $T$. Relatively high $\theta_{\mathrm{eb}}$ — especially over EAI, e.g., centred above the
Gamburtsevs — yields unstable air columns. Given that maxima in $\theta_{\mathrm{eb}}$ align with maxima in upper tropospheric temperature
$T_u$ for PI monsoons, we anticipate the occurrence of vigorous moist convection within a (narrow) convergence zone around
the $\theta_{\mathrm{eb}}$ maximum (Nie et al., 2010). The remaining region consists of (broad) areas characterised by descending air, thus
completing the overturning circulation.





Convection is then observed in regions around the edges of a high-pressure system that encompasses the entire interior of the Antarctic continent, characterised by high $\theta_{\mathrm{eb}}$ (Fig. 6). Likely because this area has a ring-like structure around the pole, surface air is advected from all around the continent (see also DJ-mean $u_{850}$ in Fig. 4). The incoming air, originating from the
Southern Ocean, carries moisture, resulting in high MSE in these areas. In order to reach the central part of the continent, air must ascend in almost every region, eventually leading to precipitation. The inflow of maritime air also moderates $T_{\mathrm{s}}$ in these areas, especially when contrasted with the lower-lying interior of the continent that extends through the valley between the Transantarctic Mountains and the Gamburtsevs towards George V Land.

At an altitude of approximately 600 hPa, this ring-like structure exhibits two cells with moisture transported towards (neg-
ative values) and away from (positive values) the continent: a large cell with negative values extending from the Weddell Sea side of the Peninsula towards Wilkes Land, and another large cell with positive values from Wilkes Land towards the Peninsula (similar to the third panel in Fig. 9, but not shown here). This implies, at height, a large-scale flow from the Atlantic sector to the Pacific sector of the continent. These cells extend vertically into higher regions of the troposphere and diminish when radiative and orbital forcing decrease. The large positive cell yields negative values at the surface throughout most areas (in-
dicating moisture transport towards the continent), except for the region around the Ross Sea: in areas such as Prydz Bay, and Oates, George V, Adélie and Marie Byrd Land, rising air due to high $T_{\mathrm{s}}$ prompts moisture advection from the Southern Ocean (resembling a sea-breeze circulation), after which this moisture is transported at height away from those regions. In Dronning Maud Land, the large-scale flow is already directed continent-wards, resulting in the transport of moisture from the Southern Ocean towards the continent.

Peak rainfall often coincides with peak surface MSE in monsoonal configurations, due to the fact that cooler and drier air predominantly results in shallow convection (Biasutti et al., 2018). MSE exhibits higher surface values during summer compared to winter, and this pattern is similarly observed for 4PIC in contrast to 2PIC/s and 2PIC/l (Fig. 6). However, this trend is not as clearly visible when considering column-integrated MSE (not shown here). Given that the spatial distribution of MSE is primarily influenced by the spatial arrangement of $q_{\mathrm{e}}$ and that regions characterised by high $q_{\mathrm{e}}$ are regions where
precipitation does not rain out significantly, the linkage between MSE and peak rainfall is not evident. Furthermore, very wet regions (Dronning Maud, Oates and George V Land, and the area spanning Prydz Bay to the Gamburtsevs) do not necessarily correspond to very warm areas, which is likely attributed to the influx of colder air originating from the Southern Ocean. Only the interior of Dronning Maud Land receives over 60% of its annual precipitation during summer, while Oates, Adélie and George V Land, and a narrow coastal strip of Dronning Maud Land also witness significant rainfall during winter. Nevertheless,
in terms of pressure distribution and convection patterns, as well as the inflow of moist air from nearby oceans, we observe striking similarities in Eocene Antarctica summer circulation with subtropical monsoonal systems as we know from present-day. Our use of different atmospheric metrics expands in this way the classification of the monsoonal nature of the Late-Eocene Antarctic atmospheric circulation compared to the monsoonal index recently used by Baatsen et al. (2024).





### 4.3 Effects of imposed ice sheets

Imposing regional ice sheets leads to regional changes in our simulations and the extent of these changes grows with the size of the ice sheet. Notably, the height and likely also the size of these ice sheets (for reference, the Dronning Maud Land ice sheet in 2PIC/li reaches a maximum elevation of approximately 4.5 km) appears to influence the atmospheric circulation, although the effects remain regional and do not extend across the entire continent. The interior of the continent, along with the coastal areas of East-Antarctica, seems to experience little to no influence from the ice sheets on Dronning Maud Land, Oates Land

and the Peninsula (Fig. 7). On the other hand, these ice sheets appear to demonstrate their capacity for self-sustenance: despite the still relatively warm and moist summer conditions, SMBs remain nearly to strongly positive across the entire ice sheet area (Fig. 10). Since ice sheets were imposed based on lowered annual surface temperatures, this result is not trivial; importantly, it suggests that there is (some) potential space, particularly in Dronning Maud and Oates Land, for expansion in the surrounding vicinity, which is also evident in Kemp Land and the high regions of the Gamburtsevs and the Transantarctic Mountains.

Other significant changes between 2PIC/li and 2PIC/l occur over the Weddell Sea region, where $T_s$ is notably higher during both summer and winter (Fig. 7). During summer, increased incoming $Q_{net}$ contributes to higher $T_s$, enhanced evaporation, and subsequently higher $q_e$, MSE and a larger cloud cover (Fig. A2, see also Fig. 8). The anti-clockwise, positive pressure anomaly over the Weddell Sea yields air flow from Coats Land encountering the Dronning Maud Land ice sheet, after which air converges and ascends, resulting in positive $P$ anomalies in that area. In winter, the air flow pattern is somewhat similar to the

pattern in 2PIC/l, crossing the continent from Dronning Maud Land towards the 180–230°E sector. Although less powerful, air still cools as it moves over and primarily along the Dronning Maud Land ice sheet towards the South Pole and the interior of the continent. The Weddell Sea thus exhibits large positive $T_s$ anomalies, but its impact on the atmospheric circulation over the continent itself appears to be limited. Nevertheless, a warmer Weddell Sea could be a potential source for increased precipitation in Dronning Maud Land and the Peninsula, which would in winter lead to increased snow and ice build-up.

Comparing these model results with available proxies is challenging, as most of these proxies are representative only for marine environments. Terrestrial proxies from South America and Australia suggest a global $T_s$ decrease of approximately 3–5 °C during the EOT (Pound and Salzmann, 2017; Colwyn and Hren, 2019), while other proxies indicate a smaller $T_s$ decrease (Lauretano et al., 2021). Methods for subglacial examination of glacial cirques suggest that the AIS likely expanded from high-elevation mountain ranges or interior massifs toward coastal areas (Bo et al., 2009; Rose et al., 2013), a process

that might have occurred well before the EOT (Gulick et al., 2017; Barr et al., 2022). Furthermore, evidence from Antarctic glaciation, based on $\delta^{18}O$ excursion events, suggests that this glaciation likely occurred within a relatively short time frame of ~500 ky (Hutchinson et al., 2018). The decrease in $T_s$ indicated by these proxies is comparable to the $T_s$ decrease observed between 2PIC/s and 2PIC/l scenarios (Table A1), which suggests that such a cooling trend could be caused by favourable orbital conditions. The regional ice sheets that could develop in response to this change in orbital forcing are of such a size that they

could sustain themselves and exert a regional influence on atmospheric circulation and resulting $T_s$ and $P$ patterns. However, these ice sheets do not appear to grow from high interior mountain ranges; instead, they seem to initiate in areas closer to the





coast. The Dronning Maud Land ice sheet particularly shows promise for expansion toward the continent's interior, contrary to the traditional assumption of growing towards the Southern Ocean.

These ice sheets exhibit SMBs in the range of 1–2 m yr$^{-1}$, potentially enabling the development of substantial ice sheets over
a couple of thousands of years. In terms of surface mass balance, a $\delta^{18}$O excursion period of ~500 ky then seems plausible to envision the growth of a continent-wide ice sheet, especially when accounting for positive feedback mechanisms that amplify ice sheet expansion. However, the type of insolation minimum we modelled here is mostly precession-based and therefore only lasts 5–10 ky. SMB would then have to remain substantially high under a much wider range of orbital parameters to yield this long excursion period of 500 ky. The limited climatic differences between the 2PIC/s and 2PIC/l scenario, nonetheless, would
suggest that this indeed could be the case.

Despite this long excursion period, we must consider the tipping point that is necessary to trigger initial glacier growth, as it would likely have taken tens of thousands of years and multiple attempts within that 500-ky range to develop glaciers and subsequent small- to moderately-sized ice sheets capable of surviving the Antarctic summer monsoon. In addition to radiative and orbital forcing, specific short-term cooling events (such as volcanic eruptions like Circum-Pacific arc flare-ups (Jicha et al.,
2009) or Tana Basin LIP volcanism (Prave et al., 2016), meteorite impacts like the Haughton (Erickson et al., 2021) or Popigai (Bottomley et al., 1997) impacts, or biologic events like *Azolla* blooms in the Arctic (Speelman et al., 2009)) could potentially trigger such a tipping point. In this context, the occurrence of two distinct $\delta^{18}$O excursion events — the precursor event and the EOIS, as mentioned above — is likely essential to initiate ice sheet growth: initially, the climate needs to cool sufficiently in order to yield a high likelihood of survival for incipient glaciers and (small- to moderately-sized) ice sheets during the summer
season. Subsequently, after a considerable period of time, conditions become favourable enough to support regional and later even continent-wide ice sheet expansion.

## 5 Conclusion

This study presented an analysis of Antarctic climate simulations at 38 Ma, representing the Late-Eocene and the Eocene-Oligocene Transition, as conducted by Baatsen et al. (2020). The primary focus was to assess the stability of an incipient AIS
under varying radiative, orbital and cryospheric forcing. The central hypothesis presented above was that an incipient AIS, comprising various regional ice sheets, could induce sufficient changes in the atmospheric circulation to moderate or even disrupt the warm and moist summer climate. With this, we aimed to shed light on the question of how the AIS emerged within a relatively short time frame of around 500 ky, given that during immediately preceding periods summer temperatures on Antarctica could easily reach 30 °C.
The climatic conditions prevailing during (the lead-up to) the EOT can be characterised as extremely seasonal, featuring a short yet intense summer period and contrasting cold winters. Precipitation patterns exhibit distinctly regional characteristics, with coastal areas being most wet during winter, while Dronning Maud, Oates, Adélie and George V Land experience largest $P$ during summer. Notably, the reduction in Antarctic summer insolation results in a weakening of this seasonality, with a pronounced change in particularly DJ-mean $T_{\mathrm{s}}$ ($-2.9$ °C for summer and $+0.8$ °C for winter). The interannual variability





within this climate is limited and aligns with results from other GCM simulations, although the CESM1.0.5 model used in this study is very wet for DJ, and although it is challenging to make a direct comparison between outcomes due to the distinct boundary conditions utilised across various models. Nonetheless, it is evident that the climatic conditions prevailing at 38 Ma would have been highly inhospitable to ice sheet growth for most of the continent, as limited snow accumulation is expected to survive the summer season.

This is also reflected in our characterisation of the monsoonal nature of the atmospheric circulation, as a warm, moist and persistent summer climate, driven by continuous incoming solar radiation as reflected in $Q_{\mathrm{net}}$. Furthermore, we observed a narrow convergence zone with moist convection around the region where $\theta_{\mathrm{eb}}$ is high. This area exhibits a ring-like structure, with moist surface air being advected from the Southern Ocean. This advection leads to high MSE values and subsequent precipitation in these regions, with pressure distribution and convection patterns similar to present-day (sub)tropical monsoonal

circulations.

Although imposed, moderately-sized, regional Antarctic ice sheets during the Late-Eocene seem to be unable to disrupt this circulation under favourable orbital conditions (characterised by low summer insolation) and moderate radiative conditions (with $p\mathrm{CO}_2$ not exceeding 2 PIC), these ice sheets appear capable of sustaining themselves and even demonstrate a propensity for modest expansion. Specific short-term cooling events (such as volcanic eruptions, meteorite impacts or biologic events such

as massive algae blooms) could potentially trigger a tipping point for widespread glaciation. Conversely, our findings emphasise a significant hysteresis effect for local and/or regional ice sheets on the Antarctic continent, suggesting that these may have been present for a substantial amount of time prior to the EOT, while not necessarily implying continent-wide glaciation.

*Code and data availability.* All model output is post-processed using Python 3.10. A selection of the model data used to generate the main figures in this paper and the necessary grids and software code is publicly available on the Utrecht University Yoda platform:

https://doi.org/10.24416/UU01-A0VMKZ (Baatsen et al., 2024). The above data are post-processed to be more accessible and only contain the variables considered in this specific work. The full data from the respective model simulations are available upon reasonable request from the authors.





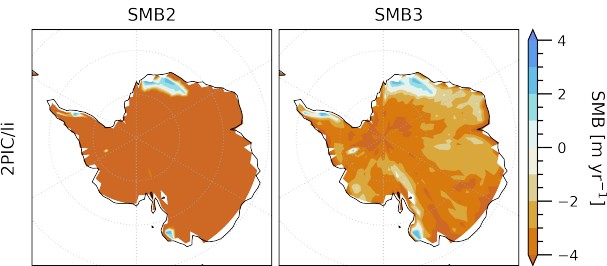

**Figure A1.** Surface mass balances of imposed regional ice sheets for 2PIC/li, calculated using two different methods: 1. a PDD method with a fixed melt rate (SMB2); and 2. a PDD method with a fixed melt rate including a variability term for $T_\mathrm{s}$ to calculate PDD (SMB3).





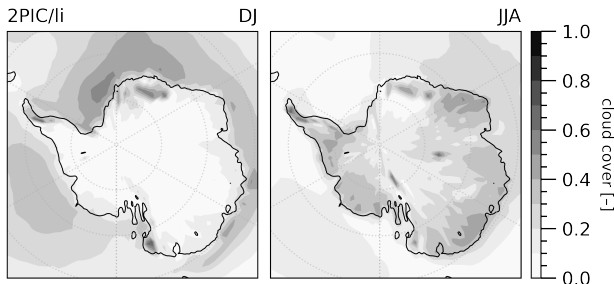

**Figure A2.** DJ-mean (left panel) and JJA-mean (right panel) 975 hPa cloud cover for the 2PIC/li scenario.





**Table A1.** Annual, DJ- and JJA-mean and (spatial) standard deviation for $T_s$ [°C] and $P$ [mm day$^{-1}$]. The values are provided for Antarctica as a whole and for EAH, EAI, EAC and WA. Values encompass all four scenarios.

| | | ALL | EAH | EAI | EAC | WA |
|---|---|---|---|---|---|---|
| **4PIC** | $T_s$ | 5.7±0.5 | 2.2±0.5 | 4.9±0.7 | 10.7±0.3 | 6.4±0.6 |
| | $T_{s,DJ}$ | 28.7±1.1 | 24.6±1.3 | 30.5±1.4 | 24.7±10.7 | 28.9±1.3 |
| | $T_{s,JJA}$ | -9.2±1.1 | -12.3±1.2 | -11.9±1.5 | 0.4±0.7 | -8.0±1.3 |
| | $P$ | 2.08±0.10 | 3.19±0.18 | 1.91±0.11 | 2.84±0.13 | 2.00±0.16 |
| | $P_{DJ}$ | 2.41±0.24 | 4.27±0.34 | 2.66±0.34 | 1.87±0.18 | 1.91±0.28 |
| | $P_{JJA}$ | 1.87±0.16 | 2.82±0.35 | 1.52±0.16 | 3.11±0.24 | 1.91±0.27 |
| **2PIC/s** | $T_s$ | 0.4±0.6 | -2.9±0.6 | -0.8±0.8 | 6.4±0.4 | 1.0±0.6 |
| | $T_{s,DJ}$ | 23.9±1.0 | 19.6±1.2 | 25.4±1.2 | 20.7±0.8 | 24.2±1.2 |
| | $T_{s,JJA}$ | -15.3±1.2 | -18.1±1.4 | -18.2±1.7 | -4.3±0.8 | -14.1±1.2 |
| | $P$ | 1.82±0.07 | 2.79±0.17 | 1.69±0.09 | 2.59±0.10 | 1.71±0.11 |
| | $P_{DJ}$ | 2.39±0.21 | 3.91±0.26 | 2.66±0.30 | 1.92±0.16 | 1.93±0.25 |
| | $P_{JJA}$ | 1.59±0.13 | 2.35±0.31 | 1.30±0.13 | 2.84±0.20 | 1.60±0.24 |
| **2PIC/l** | $T_s$ | 0.5±0.6 | -3.1±0.6 | -0.8±0.7 | 6.6±0.4 | 1.3±0.6 |
| | $T_{s,DJ}$ | 21.0±0.8 | 16.9±0.7 | 21.9±0.8 | 19.6±0.6 | 21.2±1.0 |
| | $T_{s,JJA}$ | -14.5±1.2 | -17.3±1.5 | -17.4±1.7 | -3.8±0.7 | -13.3±1.3 |
| | $P$ | 1.80±0.07 | 2.52±0.15 | 1.65±0.08 | 2.56±0.12 | 1.74±0.11 |
| | $P_{DJ}$ | 2.16±0.16 | 3.25±0.22 | 2.37±0.24 | 1.89±0.20 | 1.82±0.20 |
| | $P_{JJA}$ | 1.63±0.15 | 2.35±0.32 | 1.35±0.16 | 2.92±0.24 | 1.64±0.25 |
| **2PIC/li** | $T_s$ | 0.0±0.5 | -5.5±0.5 | -1.4±0.6 | 7.0±0.2 | 1.3±0.6 |
| | $T_{s,DJ}$ | 21.1±0.7 | 14.2±0.6 | 22.5±0.8 | 19.8±0.5 | 21.9±0.9 |
| | $T_{s,JJA}$ | -15.4±0.9 | -19.4±1.2 | -18.9±1.2 | -3.3±0.5 | 13.7±1.2 |
| | $P$ | 1.78±0.08 | 2.43±0.15 | 1.59±0.08 | 2.68±0.10 | 1.74±0.13 |
| | $P_{DJ}$ | 2.13±0.18 | 3.04±0.23 | 2.33±0.27 | 1.99±0.19 | 1.79±0.21 |
| | $P_{JJA}$ | 1.60±0.15 | 2.30±0.30 | 1.24±0.16 | 3.00±0.22 | 1.64±0.26 |





**Table A2.** Annual, DJ- and JJA-mean and (spatial) standard deviation for $Q_{\text{net}}$ [W m$^{-2}$], $\theta_{\text{eb}}$ [K] and MSE [GJ m$^{-2}$]. The values are provided for Antarctica as a whole and for EAH, EAI, EAC and WA. Values encompass all four scenarios.

| | | ALL | EAH | EAI | EAC | WA |
|---|---|---|---|---|---|---|
| **4PIC** | $Q_{\text{net}}$ | $-79.8 \pm 8.9$ | $-77.1 \pm 4.2$ | $-79.8 \pm 6.1$ | $-73.7 \pm 12.7$ | $-82.6 \pm 10.3$ |
| | $Q_{\text{net,DJ}}$ | $90.4 \pm 26.8$ | $90.1 \pm 21.4$ | $97.7 \pm 15.0$ | $64.6 \pm 38.1$ | $89.6 \pm 30.4$ |
| | $Q_{\text{net,JJA}}$ | $-151.6 \pm 18.4$ | $-150.4 \pm 5.6$ | $-155.6 \pm 3.8$ | $-139.9 \pm 29.4$ | $-151.0 \pm 25.0$ |
| | $\theta_{\text{eb}}$ | $299.8 \pm 3.6$ | $304.6 \pm 1.2$ | $300.1 \pm 2.6$ | $300.0 \pm 3.0$ | $297.4 \pm 3.3$ |
| | $\theta_{\text{eb,DJ}}$ | $331.2 \pm 7.2$ | $337.0 \pm 4.5$ | $334.8 \pm 4.5$ | $324.1 \pm 6.7$ | $326.8 \pm 6.2$ |
| | $\theta_{\text{eb,JJA}}$ | $280.6 \pm 5.9$ | $286.4 \pm 3.3$ | $279.2 \pm 5.7$ | $282.7 \pm 6.0$ | $279.2 \pm 6.5$ |
| | MSE | $46.0 \pm 6.8$ | $36.8 \pm 3.2$ | $45.3 \pm 3.3$ | $56.6 \pm 5.1$ | $47.6 \pm 6.0$ |
| | MSE$_{\text{DJ}}$ | $89.8 \pm 9.1$ | $78.1 \pm 6.8$ | $93.2 \pm 6.0$ | $100.0 \pm 6.1$ | $87.9 \pm 7.9$ |
| | MSE$_{\text{JJA}}$ | $23.9 \pm 6.0$ | $17.6 \pm 2.3$ | $22.2 \pm 3.1$ | $32.2 \pm 5.9$ | $26.0 \pm 6.0$ |
| **2PIC/s** | $Q_{\text{net}}$ | $-79.4 \pm 9.4$ | $-77.0 \pm 4.5$ | $-79.6 \pm 6.2$ | $-73.5 \pm 13.3$ | $-81.8 \pm 11.4$ |
| | $Q_{\text{net,DJ}}$ | $83.8 \pm 28.3$ | $79.7 \pm 26.5$ | $91.6 \pm 18.3$ | $58.9 \pm 38.1$ | $83.8 \pm 30.5$ |
| | $Q_{\text{net,JJA}}$ | $-147.8 \pm 19.1$ | $-147.0 \pm 4.8$ | $-151.7 \pm 3.5$ | $-136.8 \pm 29.6$ | $-146.9 \pm 26.5$ |
| | $\theta_{\text{eb}}$ | $292.9 \pm 3.9$ | $298.0 \pm 1.4$ | $293.2 \pm 2.9$ | $293.4 \pm 3.2$ | $290.3 \pm 3.6$ |
| | $\theta_{\text{eb,DJ}}$ | $323.1 \pm 6.5$ | $328.4 \pm 4.1$ | $326.4 \pm 3.9$ | $316.9 \pm 6.3$ | $318.9 \pm 5.7$ |
| | $\theta_{\text{eb,JJA}}$ | $274.5 \pm 6.2$ | $280.4 \pm 3.6$ | $273.0 \pm 5.0$ | $276.7 \pm 6.2$ | $273.1 \pm 6.7$ |
| | MSE | $34.6 \pm 5.5$ | $27.5 \pm 2.6$ | $34.1 \pm 2.9$ | $43.6 \pm 4.7$ | $35.4 \pm 4.8$ |
| | MSE$_{\text{DJ}}$ | $70.1 \pm 7.6$ | $60.6 \pm 5.6$ | $72.8 \pm 4.5$ | $80.0 \pm 5.1$ | $68.2 \pm 6.5$ |
| | MSE$_{\text{JJA}}$ | $17.1 \pm 4.8$ | $12.3 \pm 1.8$ | $15.8 \pm 2.6$ | $23.9 \pm 4.9$ | $18.7 \pm 4.7$ |
| **2PIC/l** | $Q_{\text{net}}$ | $-84.5 \pm 9.9$ | $-82.3 \pm 4.3$ | $-84.6 \pm 6.9$ | $-76.5 \pm 14.0$ | $-87.6 \pm 11.3$ |
| | $Q_{\text{net,DJ}}$ | $26.8 \pm 19.3$ | $27.2 \pm 17.8$ | $33.0 \pm 11.6$ | $10.5 \pm 27.9$ | $24.4 \pm 20.6$ |
| | $Q_{\text{net,JJA}}$ | $-146.9 \pm 19.1$ | $-145.6 \pm 4.7$ | $-150.4 \pm 3.6$ | $-134.6 \pm 28.9$ | $-147.0 \pm 26.7$ |
| | $\theta_{\text{eb}}$ | $292.9 \pm 3.7$ | $297.3 \pm 1.3$ | $292.9 \pm 2.8$ | $293.8 \pm 3.3$ | $290.7 \pm 3.5$ |
| | $\theta_{\text{eb,DJ}}$ | $319.5 \pm 4.9$ | $323.7 \pm 2.7$ | $322.3 \pm 2.7$ | $315.3 \pm 4.8$ | $315.9 \pm 4.2$ |
| | $\theta_{\text{eb,JJA}}$ | $275.1 \pm 6.1$ | $281.0 \pm 3.3$ | $273.7 \pm 4.8$ | $277.4 \pm 5.8$ | $273.6 \pm 6.7$ |
| | MSE | $33.1 \pm 5.9$ | $25.4 \pm 2.5$ | $32.1 \pm 3.0$ | $42.2 \pm 4.9$ | $34.7 \pm 5.1$ |
| | MSE$_{\text{DJ}}$ | $62.5 \pm 7.4$ | $51.8 \pm 4.6$ | $63.8 \pm 4.1$ | $74.1 \pm 4.6$ | $62.1 \pm 5.8$ |
| | MSE$_{\text{JJA}}$ | $18.0 \pm 4.9$ | $13.0 \pm 1.7$ | $16.6 \pm 2.5$ | $24.9 \pm 4.8$ | $19.7 \pm 4.8$ |
| **2PIC/li** | $Q_{\text{net}}$ | $-84.4 \pm 10.2$ | $-85.5 \pm 9.4$ | $-84.2 \pm 6.3$ | $-77.6 \pm 14.6$ | $-86.0 \pm 11.7$ |
| | $Q_{\text{net,DJ}}$ | $24.5 \pm 26.4$ | $16.9 \pm 38.5$ | $31.4 \pm 17.2$ | $4.9 \pm 34.8$ | $25.5 \pm 22.1$ |
| | $Q_{\text{net,JJA}}$ | $-145.4 \pm 19.6$ | $-145.0 \pm 7.1$ | $-149.2 \pm 3.5$ | $-132.8 \pm 29.5$ | $-144.9 \pm 27.2$ |
| | $\theta_{\text{eb}}$ | $293.1 \pm 4.2$ | $297.9 \pm 2.2$ | $292.9 \pm 3.5$ | $294.6 \pm 3.8$ | $290.8 \pm 3.9$ |
| | $\theta_{\text{eb,DJ}}$ | $320.0 \pm 4.8$ | $323.8 \pm 3.2$ | $322.8 \pm 2.7$ | $315.9 \pm 4.7$ | $316.5 \pm 4.0$ |
| | $\theta_{\text{eb,JJA}}$ | $275.2 \pm 7.3$ | $281.8 \pm 5.5$ | $273.3 \pm 6.1$ | $278.4 \pm 7.0$ | $273.6 \pm 7.5$ |
| | MSE | $33.0 \pm 6.9$ | $24.2 \pm 5.2$ | $32.0 \pm 3.9$ | $42.7 \pm 6.2$ | $35.0 \pm 5.4$ |
| | MSE$_{\text{DJ}}$ | $62.8 \pm 9.4$ | $49.7 \pm 10.5$ | $64.4 \pm 5.5$ | $75.0 \pm 7.4$ | $62.9 \pm 6.1$ |
| | MSE$_{\text{JJA}}$ | $17.8 \pm 5.6$ | $12.2 \pm 2.8$ | $16.0 \pm 3.3$ | $25.1 \pm 5.9$ | $20.0 \pm 5.2$ |





**Table A3.** Boundary conditions for one model used in EoMIP and for four models used in DeepMIP. Resolution is provided for the atmosphere only, and $p\text{CO}_2$ is given in PI concentration (i.e., not equal to PIC).

| Model | Resolution | $p\text{CO}_2$ | Geography |
|---|---|---|---|
| CCSM3 | 3.75°×3.7°×26 | 4× | Sewall et al. (2000) |
| CESM1.2 | 1.9°×1.9°×30 | 3× | Herold et al. (2014) |
| HadCM3 | 2.5°×3.8°×19 | 3× | Herold et al. (2014) |
| IPSL | 1.9°×2.5°×39 | 3× | Herold et al. (2014) |
| NorESM | 2.0°×2.0°×36 | 4× | Baatsen et al. (2016) |



*Author contributions.* DV conceived the idea for this study, after which DV, AvdH and MB contributed to the conceptualisation of the narrative and analyses needed. MB and AvdH designed the 4PIC, 2PIC/s and 2PIC/l model simulations; DV and MB designed the 2PIC/li model simulation. DV post-processed the data, conducted the analyses and constructed the figures. DV set up a first draft of the manuscript. MB and AvdH provided revisions to the manuscript, after which DV wrote its final version.

*Competing interests.* The authors declare that they have no conflict of interest.

*Acknowledgements.* The authors thank Heiko Goelzer for assisting with the ISM simulation, and Michael Kliphuis for assisting with all model simulations and management of the output data. This work was carried out under the program of the Netherlands Earth System Science Centre (NESSC), financially supported by the Ministry of Education, Culture and Science (OCW, grant 024.002.001). Simulations were performed at the SURFsara Dutch national computing facilities and were sponsored by NWO-ENW (Dutch Research Council, Exact and Nature Sciences) under the projects 17189 and 2020.022. The work of AvdH was also funded by NWO through the Vici project 'Interacting climate tipping elements: When does tipping cause tipping?' (project VI.C.202.081).



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
