# Peer review of "Sustainability of regional Antarctic ice sheets under late-Eocene seasonal atmospheric conditions"

_Climate of the Past, 2024_

## Author Response (AR1)

**Author's response to peer review**

D.H.A. Vermeulen, M.L.J. Baatsen and A.S. von der Heydt

**Changes in this new version of the manuscript**

General
- The title of the paper has been changed.
- We updated abstract so that it resembles changes in the main text.
- We updated section headings to better resemble contents of the respective section.
- We updated the conclusion so that it resembles changes in the main text.
- We fixed typos and sentence structures.

Introduction
- We moved info about the two phases in d18O excursion, so that is connected to our other explanation about the d18O excursion.
- We updated the paragraph about other model studies.

Methods
- We included a panel of Late-Eocene Antarctic points of interest in Figure 1.

Results
- We limited the amount of words spent on the analysis of the different Antarctic regions in all three sections.
- We limited our results on the Weddell Sea.
- We removed our results on the vertical structure of the atmosphere.

Discussion
- We limited our discussion of the model scenarios already discussed in Baatsen et al. 2024
- We removed our comparison of the Late-Eocene Antarctic regime with present-day (sub)tropical monsoonal climates.
- We removed our discussion of the choice of atmospheric model used in DeepMIP.
- We removed our discussion of the vertical structure of the atmosphere.
- We added a paragraph about another study describing Antarctic monsoons during the Eocene.
- We added an analysis of the sensitivity of the SMB to the different components of the various SMB methods used.
- We removed our discussion of the tipping point necessary to induce glacier and ice sheet growth.

**Response to review 1**

Vermeulen et al. conduct a ice sheet model and climate model simulation, and analyze a set of existing climate model simulations, to investigate incipient Antarctic Ice Sheet (AIS) stability under a warm and monsoonal late Eocene climate which is thought to be inhospitable to ice sheet growth. They hypothesize that a moderate-size AIS could have disrupted this inhospitable climate, thereby triggering a feedback mechanism for glacial expansion at the EOT. They test this hypothesis and find that the prescribed ice sheets do not significantly disrupt this warm monsoonal climate, but their ice sheets still demonstrate the potential to self-sustain and even grow (i.e., they calculate positive surface mass balance, SMB).

The manuscript has clear prose and is well organized. The hypothesis is interesting, and their investigation contributes to this scientific field. The impact of the imposed ice sheets on the modeled climate is extensively described and discussed, supporting their first finding (that the imposition of these ice sheets into the model does not significantly disrupt the preexisting climate).
*AR: Thank you for your positive evaluation.*

However, I thought that the second finding (that these ice sheets are able to self-sustain and even grow) was missing an important discussion: what factor is dictating these positive SMB calculations? Lots of moisture? The calculation type (e.g., a PDD scheme or energy balance scheme? And does this relate to whether summer intensity vs summer duration is more impactful on ice sheet stability? E.g., Raymo & Huybers, 2008). The authors state many times that the monsoonal/seasonal late Eocene warm climate is inhospitable to ice growth due to limited summer snow accumulation -- but in the end, they find that this climate is not, in fact, inhospitable to ice growth. So I'm curious - what was that original assumption based on? Is it simply wrong? The authors don't go back and reassess that claim.
*AR: That's a good point; we will add this to the final version of the manuscript.*

I must admit that I mostly think about paleo ice sheet stability and I lack expertise in atmospheric circulation dynamics. Given that caveat, I wonder if portions of the Results & Discussions sections could be shortened or streamlined. I was often confused about the relevance of the detailed analyses/descriptions of model climate dynamics (although this may have more to do with lack of expertise rather than the quality of the manuscript). I very much appreciated sentences that spelled out the relevancy of the various analyses (for example, L319, "This is a crucial part of the Antarctic Eocene climate, as the circulation pattern basically redirects the storm track... resulting in high precipitation over DML..").
*AR: We agree. This is also mentioned in RC2, so we will go over the Results and Discussion sections again to make those more concise.*

I also wondered fairly frequently while reading this manuscript about 'false precision', i.e., how much of these details around wind direction and seasonality are dictated by uncertainties in the model or boundary conditions, rather than a robust paleo-climate feature. The authors do address this a bit in Sect 4.1 by comparing their results to other models, though; I simply mention this as food for thought.
*AR: That indeed is a good question. In our opinion, large seasonality is a robust feature that is also visible in other models, as indeed is discussed in Sect. 4.1. We did not extensively compare (surface or 850 hPa) pressure to other models, but qualitatively it makes sense that during JJA a (thermal) high pressure area is formed over the (East-)Antarctic continent and a (thermal) low pressure area over the Weddell*

*Sea. Furthermore, this dipole is a robust feature in all of our scenarios. Of course, this could be model-dependent, so we will elaborate a bit more on this nuance in Sect. 4.1.*

In sum, I recommend this paper for publication, but found the ice sheet analyses a bit underdeveloped relative to the stated premise of the paper. A few small comments:

- Abstract: L13-16 is restating L7? And it seems like that sentence is stating  existing information - from Baatsen et al 2024 - but then the next sentence is a finding of this work?
  *AR: We agree and we will correct this.*
- Some parentheses are unnecessary and break up the flow of the narrative (for example, L51, L64, L81, etc.)
  *AR: We will check again where parentheses disturb the flow of reading.*
- L84: I found the 'ab- and presence' to be jarring, similarly 'topo-and geography'
  *AR: We will correct this.*
- L299: Here and elsewhere, re: late Eocene seasonality/monsoonal conditions, I'm not always sure what is a reference to Baatsen et al 2024 versus new findings from this study.
  *AR: Good point, we will make this clearer.*

**Response to review 2**
This manuscript performed several paleoclimate simulations to study the Antarctic climate conditions during the EOT with focuses on the role of Earth's orbital forcing and the impact from incipient ice sheets. The authors' simulations suggest strong seasonal and monsoonal climate over the Antarctica with intense summer warming that prevents large-scale snow accumulation. When imposing the ice sheets derived from offline ice sheet model, the authors find regional positive surface mass balance, indicating the potential of growth of the incipient ice sheets. The authors' results suggest the potential for the existence of substantial volume of ice before the EOT. The research topic is of high relevance and fits nicely with Climate of the Past. I enjoyed reading the abstract and the introduction very much.
*AR: Thank you for your positive evaluation of the abstract and the introduction.*

I think the manuscript need revision to better highlight its key points and novelty before it can be published in Climate of the Past. Below are my major comments on Results and Discussion sections, along with a few minor comments and editing suggestions.

Major comments:
1. I think the Results and Discussion sections could be improved to highlight the most important findings and novelty from the present study. In its current form, the Results and Discussion simply lists many detailed descriptions of simulation results without sufficient summary of the main points of each subsection/paragraph, making it very difficult for the readers to follow.
   *AR: We agree, and since this is also mentioned in RC1, we will make the Result and Discussion sections more concise. Furthermore, we will add summary paragraphs at the end of each subsection in the Results section to highlight the most important findings.*
2. At places, the manuscript is very descriptive and lacks in-depth analysis/understanding on the results. For example, which processes drive the SMB? If the incipient ice sheets were imposed over other regions (e.g., more interior), should we expect similar SMB (based on our understanding of the physical processes)?
   *AR: This point is also mentioned in RC1. Specifically for the section about the ice sheets and SMB we will add a clearer interpretation, but we will go over the whole Discussion section again to make it less descriptive.*

Minor comments:
1. Title needs to be revised. The manuscript is focusing on Antarctica, not the global climate. I suggest the authors revise the title to reflect the regional focus of the present manuscript.
   AR: We interpret 'on Antarctica' at the end of the title as applying to the title as a whole, but apparently this causes confusion. We will think of a better way of putting emphasis on the regional scale of the study.
2. Line 17: check the writing of "is high is shown".
   *AR: We will correct this.*
3. Line 97: the better performance of the newer models in Lunt et al. 2021 is not attributed to the "higher resolution". Instead, it may be attributable to the improved model physics (e.g., Zhu, Poulsen, & Tierney, 2019).
   *AR: Indeed, thank you for pointing this out. We will correct this.*
4. Line 118: −1.8 °C (not 1.8 °C).
   *AR: We will correct this.*

5. Table 1: check the units of obliquity and precession.
   AR: Indeed, this should not be [°C] but only [°]. Thank you for your keen eye.
6. Subsection title "Ice sheet Stability" could be revised to, for example, "Ice sheet sustainability".
   *AR: We agree the topic is more on sustainability than stability, so we will correct this.*
7. Line 338: delete one redundant ")"
   *AR: We will correct this.*
8. The use of many region names, such as Dronning Maud Land, Marie Byrd Land, may be hard for some readers who are not familiar with the Antarctica geology to follow. Please consider labeling names of key regions in Figure 1.
   *AR: We were under the impression that this was already the case, but we used an old version of Figure 1. Thank you for pointing out; we will include the newer version.*

---

## Author Response (AR2)

**Author's response to peer review 2**
D.H.A. Vermeulen, M.L.J. Baatsen and A.S. von der Heydt

**Response to reviewer 1**
- Line 211: The word 'effects' should be changed to 'affects'.
  *AR: corrected.*

- Figure 10: According to SMB3 and SMB4, both temperature and precipitation contributed to the increase of SMB for the Dronning Maud Land ice sheet, correct? This analysis does not rule out the possibility that other regions may develop ice sheets if incepting ice sheets are seeded, is that correct?
  *AR: that's not correct per se. We have looked specifically at places where our ISM forms ice sheets given a strong perturbation of the system — that has been the basis for our seeding. We cannot exclude other areas / regions (especially when they are smaller / more subtle than the areas we considered), but our method already implies a large forcing. Based on these simulations, large glaciated regions other than the ones we analysed are therefore rather unlikely to occur.*

**Response to editor review**
- Sustainability vs stability: This popped up in response to comments by one of the reviewers, to the point where the title now features the term "sustainability" and the term is also used in a section heading. However, throughout the manuscript the term ice-sheet "stability" is still used frequently: this leads to potential terminological confusion for readers. It may be useful to explicitly state somewhere (introduction?) what you mean by these two different terms, and then carefully go through the manuscript to confirm that the terms have been applied as you intended.
  *AR: thank you, we agree that we must choose one term. We picked sustainability, and changed it throughout the manuscript.*

- Throughout: "early" "middle" and "late" modifiers for the Eocene are not formal chronostatigraphic terms and thus should not have leading uppercase letters.
  *AR: corrected.*

- Line 31 and first paragraph in general: What is a "balanced" d18O value? Can you describe the excursions more directly in terms of direction and magnitude?
  *AR: changed this line to include indeed the direction and magnitude of the two excursion phases.*

- Lines 31/32: "…it is commonly interpreted as…" Worth slight revision here to make it clear that "it" refers to the EOT.
  *AR: corrected.*

- Line 36: re: point 2 about "level fall in Antarctic coastal sediments"… Can you rephrase for increased clarity/nuance? My very quick reading of the two citations suggests that it's the combination of isostatically-induced relative sea-level *rise* along Antarctic coast close to the new ice sheet—with far-field and core evidence for sea-level drop elsewhere—that is the collective evidence for ice-sheet formation.

*AR: indeed, that's correct. We changed it in the manuscript.*

- Lines 51-53: Consider using the new CenCO2PIP compilation of Cenozoic CO2: https://www.science.org/doi/10.1126/science.adi5177.
  *AR: thank you for the suggestion. We included the reference and changed the values accordingly.*

- Fig 7: As suggested by the reviewer of the revised ms, please re-examine the caption and figure (units on color ramps, etc).
  *AR: units on the color ramp were already included, but we changed the caption to match the panels shown.*